# A genome-wide association study identifies *EYA2* as a contributing gene for diabetic retinopathy in type 2 diabetes
Tengda Cai [1], Qi Pan[1], Yiwen Tao[1], Charvi Nangia[2], Aravind L. Rajendrakumar[3], Yunyan Ye[4], Tania Dottorini[5], Mainul Haque[6], Colin NA Palmer[2], Yongqing Shao[7] ✉ & Weihua Meng [1,2,8] ✉

## Abstract

**Background:** Diabetic retinopathy (DR) is a complication of diabetes that affects the eyes. This study aims to identify the genetic variants associated with DR in type 2 diabetes (T2D) patients from the UK Biobank cohort ($n = 16,988$).
**Methods:** We conducted a genome-wide association study (GWAS) of DR and integrated genomic results with multi-omics data to identify and prioritize susceptibility variants and genes. The findings are set to undergo validation in four replication cohorts.
**Results:** Here we show that the lead SNP rs6066146 in *EYA2* reaches genome-wide significance ($p = 4.21 \times 10^{-8}$) and is replicated in three independent cohorts. The SNP-based heritability for DR is estimated at 14.6% (standard deviation: 0.11). Colocalization analysis at the *EYA2* locus suggests moderate colocalization (PP.H4 = 0.553) alongside distinct association signals for DR and T2D, and cis-Mendelian randomization (MR) within the *EYA2* region provides gene-centric evidence that T2D exerts a significant causal effect on DR. Exploratory multivariable MR identifies proinsulin as a significant mediator of T2D on DR, which may partly account for the moderate evidence for colocalization. Tissue expression, chromatin interaction, and transcriptome-wide association analyses point to the spleen, while gene set analysis identifies B-cell pathways. Together, these convergent signals suggest that splenic B-cell abundance could serve as a predictive marker for DR risk.
**Conclusions:** Our study demonstrates a genomic risk locus in gene *EYA2* associated with DR in type 2 diabetes, which offers deeper insights into broader trait architecture on DR.

## Plain language summary

Diabetic retinopathy (DR) is an eye disease caused by diabetes that can lead to vision loss. We studied about 17,000 people with type 2 diabetes and compared the DNA of those with and without DR. A genome-wide association analysis is a method that scans the entire genome for markers associated with a specific condition. By scanning their entire genomes and combining this with other biological information, we found a strong risk signal at the *EYA2* gene on chromosome 20 that impacts DR. This gene has also been linked to type 2 diabetes itself. Our findings suggest that *EYA2* and its pathways could be promising targets for future treatments that might help protect against both diabetes and its retinal complications.

Diabetic retinopathy (DR), a disease influenced by multiple genes, stands as one of the most frequent ocular disorders in individuals who have diabetes. Around 35% of the global diabetic population is affected by DR, which is a leading cause of blindness in the workforce-aged demographic, despite the implementation of screening and treatments[1,2]. Effective interventions, including strict control of blood glucose and blood pressure, lipid-lowering treatments, and laser therapies, can help slow the progression of the disease and preserve vision[3]. As the global population ages and diabetes becomes more prevalent, it is anticipated that instances of DR will rise. Early diagnosis and appropriate treatment can prevent visual impairment and avoid the risk of blindness.

In diabetic patients, DR is a common microvascular complication and is associated with an increased risk of life-threatening systemic vascular complications[4]. Hyperglycemia leads to changes in the retina, causing vascular permeability, inflammation, fluid leakage, and ischemia[5]. Hyperglycemia and dyslipidemia are thought to disturb the homeostasis of the retina

[1]Nottingham Ningbo China Beacons of Excellence Research and Innovation Institute, University of Nottingham Ningbo China, Ningbo, China. [2]Division of Population Health and Genomics, School of Medicine, University of Dundee, Dundee, UK. [3]Institute for Health Equity Research, Icahn School of Medicine at Mount Sinai, New York, NY, USA. [4]Department of Ophthalmology, Lihuili Hospital affiliated with Ningbo University, Ningbo, China. [5]Department of Infectious diseases, School of Immunology and Microbial Sciences, King's College London, London, UK. [6]School of Mathematical Sciences, University of Nottingham Ningbo China, Ningbo, China. [7]Department of Ophthalmology, The Affiliated Ningbo Eye Hospital of Wenzhou Medical University, Ningbo, China. [8]Center for Public Health, Faculty of Medicine, Health and Life Sciences, School of Medicine, Dentistry and Biomedical Sciences, Queen's University Belfast, Belfast, UK.
✉e-mail: sunniesyq@163.com; weihua.meng@nottingham.edu.cn; w.meng@dundee.ac.uk

by inducing inflammatory responses in retinal tissue, including oxidative stress[6]. Epidemiologic studies have suggested multiple risk factors associated with the development and progression of DR, including higher blood glucose, hypertension, higher HbA1c, albuminuria, longer duration of diabetes (DoD), and higher blood urea concentration[7–9].

Studies have shown that among patients with 20 years of diabetes, almost all type 1 diabetes (T1D) patients and 58% of type 2 diabetes (T2D) patients have signs of DR[10]. Both twin studies and family studies have documented that the susceptibility to DR is genetically inherited in patients with T1D and T2D[11,12]. In individuals, the retinal disorder progresses from non-proliferative diabetes to vision-threatening proliferative diabetes, characterized by the growth of abnormal new blood vessels in the retina[13]. The new blood vessels and the ensuing contraction of fibrous tissue can deform the retina, leading to traction retinal detachment and severe vision loss[13]. Notably, the genetic influence is more pronounced in the severe forms of DR, with the heritability factor rising from 18% in non-PDR stages as sibling models to a substantial 50% in PDR cases[14,15].

Common variants in genes involved in processes such as glucose metabolism, lipid metabolism, inflammation, vascular regulation and cell communication have been studied for their association with DR, but results have been inconsistent[16,17]. Recently, various genome-wide association studies (GWAS) have identified numerous potential genetic loci linked to DR[3,7,18–22]. However, the reported GWAS loci for DR exhibit variations and lack broad and coherent replicability, hindering more in-depth analysis of DR. Insufficient grasp of DR's genetic aspects impedes the discovery of new biological routes for intervention, consequently escalating the healthcare expenses related to DR. To better understand the genetic mechanisms related to DR in T2D patients, we carried out a GWAS using the UK Biobank (UKB) cohort.

In this study, we identify rs6066146, located within the *EYA2* gene on chromosome 20, as a primary risk locus for the DR. Integrative multi-omics analyses further confirm that this variant is associated not only with DR but also with susceptibility to T2D. Multivariable Mendelian randomization (MVMR) findings suggest that proinsulin may mediate the effect of T2D on DR.

## Methods

### Information on cohorts and patients
The UKB serves as a large-scale biomedical database and research resource, offering comprehensive genetic and phenotypic data on roughly half a million UK participants aged between 40 and 69 across England, Scotland, and Wales. For additional details regarding the UKB cohort, please refer to the website www.ukbiobank.ac.uk. The research adhered to the principles outlined in the Declaration of Helsinki. After explaining the nature of the survey to all participants, the UKB Project obtained informed consent. Ethical approval for this research was appropriately secured from the National Health Service National Research Ethics Service (reference 11/NW/0382).

In this research, the genetic data came from the UKB cohort. They used a standardized process for DNA extraction and quality control (QC), in which detailed methods can be found at https://biobank.ctsu.ox.ac.uk/crystal/ukb/docs/genotyping_sample_workflow.pdf. An overview of our study design is presented in Fig. 1. We defined cases and controls by phenotype filtering using the following UKB field ID codes. Only White British participants (field ID: 21000) were included. All samples were drawn from individuals with T2D, identified by a doctor's diagnosis (field ID: 2443) and a "No" response to initiating insulin within one year (field ID: 2986). Among T2D patients, DR cases were those with ICD-10 code "H360" (field ID: 41202), whereas controls showed no record of this code. The numbers and clinical characteristics of cases and controls of the GWAS are shown in Supplementary Table 1.

The following four GWAS cohorts were used for replication: 1. GoDARTS (Diabetes Audit and Research in Tayside Scotland)[23]. 2. FinnGen[24]. 3. African American and European ancestry[25]. Supplementary Table 2 shows the cohorts' information.

### Quality control
Before GWAS, during the QC step, SNPs with imputation INFO scores <0.7 and minor allele frequency (MAF) < 1% were removed by using GCTA. In addition, SNPs deviating from the Hardy-Weinberg equilibrium (HWE < 0.001) and samples with high missing call rates (>0.1) were excluded by

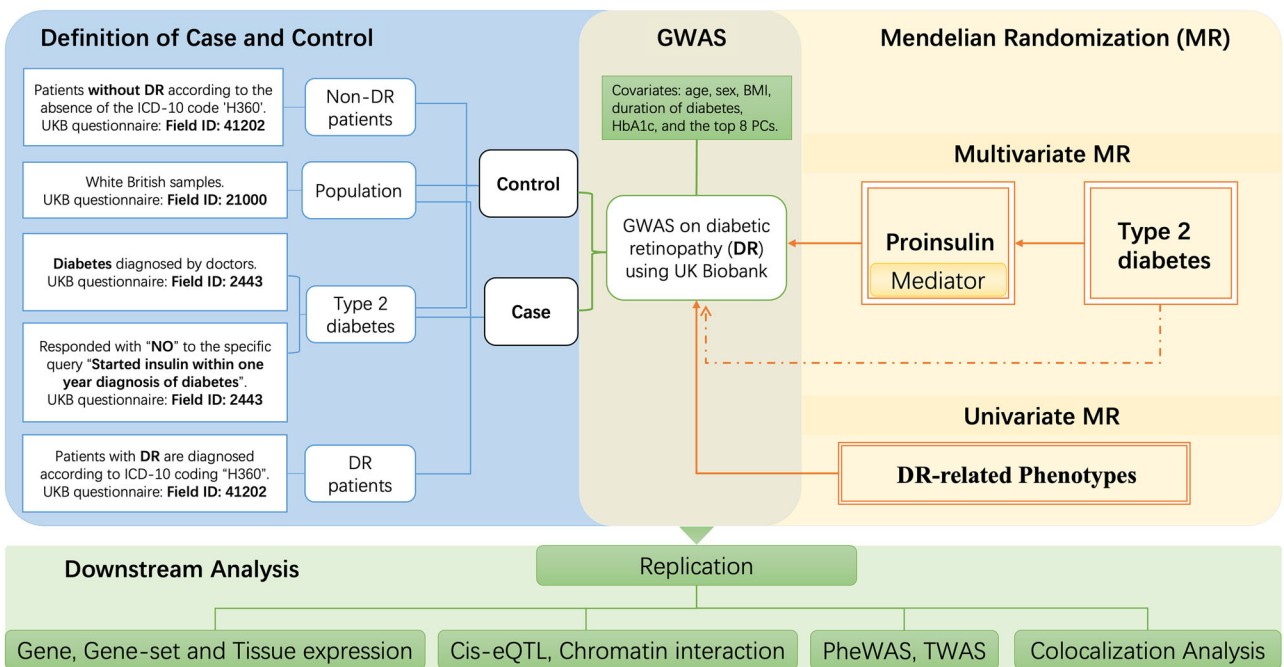

**Fig. 1 | Overview of the study.** The case and control sample definitions used in the GWAS are illustrated in the blue region based on the UK Biobank questionnaire. Case samples were defined as patients with diabetic retinopathy (DR) within the white British population with type 2 diabetes, whereas control samples were defined as non-DR patients within the same population. The yellow region represents the Mendelian Randomization (MR) analysis for DR, including multivariate and univariate MR. The green section in the middle and below outlines the GWAS and downstream analyses, including gene-based analysis, eQTL and chromatin interaction analysis, PheWAS, TWAS, colocalization, etc.

using Plink v1.90. SNPs situated on the X and Y chromosomes and mitochondrial SNPs were eliminated from consideration.

## Statistics and reproducibility

We employed the genome-wide association methods utilizing the fastGWA-GLMM (generalized linear mixed model) tool incorporated in Genome-Wide Complex Trait Analysis (GCTA v1.93.3)[26]. In the GWAS, covariates incorporated into the model were age, sex, BMI, duration of diabetes (DoD), and the top 8 genetic principal components. SNPs were considered to reach genome-wide association significance if they had a $p$ value less than $5 \times 10^{-8}$. The sex difference between cases and controls was compared using two-sided chi-square testing, whereas age, BMI, and DoD were compared by conducting independent two-sided t-tests with R v4.0.3. Moreover, SNP-based heritability was calculated using the Baseline LD model[27,28] in SumHer[29,30].

Our main replication strategy involved identifying in each cohort (GoDARTS, FinnGen, African American ancestry, and European ancestry) the SNPs corresponding to the genome-wide significant variants ($p < 5 \times 10^{-8}$) from our UKB GWAS and extracting their $p$ values. To ensure consistency, all alleles were harmonized to the UKB reference strand. Finally, we looked up the top DR-associated SNPs reported in these independent cohorts within our UKB summary statistics. A genome-wide significant SNP was considered replicated if it showed nominal evidence of association in an independent cohort ($p < 0.05$).

## Functional annotation and data visualization

Using the GRCh37 reference genome as a basis for SNP annotation, we employed FUMA v1.5.2, a web-based platform for annotating, prioritizing, visualizing, and interpreting GWAS findings[31], to perform SNP functional annotations.

For downstream analyses, we employed MAGMA v1.08, integrated within FUMA, to (i) perform gene analysis by aggregating SNP-level summary statistics at the gene level and evaluating the joint associations of all SNPs within each gene; (ii) to conduct gene-set analyses by mapping SNPs to genes using default naive positional mapping-assigning SNPs within 10 kb upstream or downstream of the annotated gene body. Gene-level p-values were then computed and tested across 10,678 MSigDB v6.2 gene sets (4761 curated sets + 5917 GO terms) with Bonferroni correction; and (iii) carry out tissue expression analyses using data from the Genotype-Tissue Expression (GTEx) project (https://www.gtexportal.org/home/) across 30 general and 54 specific tissue types to explore the relationship between tissue-specific high expression levels of genes and genetic associations in DR.

We annotated all lead and proxy SNPs with ANNOVAR within FUMA. Positional mapping then assigned each SNP to any gene whose coding region, including a 10 kb upstream and downstream flanking window, overlapped the SNP's genomic coordinate. Using this ANNOVAR-derived SNP list, we next carried out cis-eQTL mapping with GTEx v7 and v8, testing every SNP within ±1 Mb of a gene for expression association and retaining SNP-gene pairs that met a False Discovery Rate (FDR) < 0.05. For chromatin interaction mapping, we used FUMA's built-in interaction datasets (e.g., Hi-C, ChIA-PET) with a default FDR < $1 \times 10^{-6}$. Promoter regions were defined as 250 bp upstream to 500 bp downstream of each transcription start site. Any SNP (or LD proxy) involved in a significant loop that connected to a genomic fragment overlapping a promoter was assigned to that gene.

Using FUMA's GENE2FUNC functionality, hypergeometric tests were performed to assess the enrichment of our genes from gene mapping against curated functional gene sets. All genes (converted to Entrez IDs) served as the background and overlaps with each predefined set (e.g., GO terms) were tested for significance. Raw $p$ values were adjusted by Benjamini–Hochberg FDR within each category, with FDR < 0.05 deemed significant.

All parameters used in FUMA are provided in Supplementary Table 3. Q–Q plots, genomic inflation factors ($\lambda$), and regional association plots were generated with LocusZoom[32] to further characterize the genomic architecture of the trait. Summary statistics Manhattan plots were created using the R package CMplot.

## Evaluation of ocular tissue expression and phenome-wide association study (PheWAS)

The Human Protein Atlas[33] (https://www.proteinatlas.org/) was used to examine tissue-type-specific and cell-type-specific RNA profiles of candidate genes in ocular tissues. Both bulk tissue expression data (FANTOM) and single-cell expression data (Single Cell Type and Tabula Sapiens) were analyzed to delineate cell-type–specific patterns in the retina. Gene expression levels were extracted as normalized tags per million (nTPM) or equivalent standardized metrics.

We examined the PheWAS profile of each lead variant using the Type 2 Diabetes Knowledge Portal's PheWAS tool, which provides a consolidated $p$ value via an overlap-aware, fixed-effect meta-analysis[34]. Briefly, summary statistics are harmonized across cohorts, common variants (MAF > 0.05) are meta-analyzed within each ancestry with covariance adjustment for sample overlap. These ancestry-stratified findings are then merged through a second fixed-effects meta-analysis to produce trans-ethnic effect sizes (BETA for continuous traits, OR for binary) and $p$ values. We considered associations with $p < 0.05$ as nominally significant.

## Transcriptome-wide association studies (TWAS)

We employed Transcriptome-Wide Association Studies (TWAS)[35] (http://gusevlab.org/projects/fusion/) to assess the impact of gene expression modulated by genetic variants on DR risk, using expression weights derived from the GTEx v7 tissue panel and 1000 Genomes Phase 3 European (1000 G.EUR) haplotypes as our LD reference. We ran FUSION's default suite of predictive models (BLUP, LASSO, etc.) and selected the best-performing weight set per gene based on cross-validation $R^2$. Rather than testing 48 GTEx tissues, our analysis was limited a priori to the specific tissues identified by prior tissue-expression and cis-eQTL screening, and only tests for gene modules harboring significant SNPs. Because this analysis was a priori-driven, we report nominal significance at $p < 0.05$ without additional multiple-testing correction. TWAS integrated SNP-gene expression associations from eQTL with SNP-disease associations from GWAS summary statistics.

## Mendelian randomization (MR)

We conducted MR analyses to assess potential causal relationships between DR and T2D-related traits. Specifically, we incorporated genetic instruments for type 2 diabetes (test samples $n = 41$) and proinsulin levels ($n = 1$) given that proinsulin alterations in early T2D precipitate DR[36], and for cataract ($n = 2$), glaucoma ($n = 2$), and eyelid disorders ($n = 5$) since these ocular diseases are related to DR[37,38]. These analyses were performed using the TwoSampleMR[39] package in R, with all summary statistics derived from the European GWAS and accessed via the IEU OpenGWAS database[40] (see Supplementary Table 4 for details). For each exposure, we selected independent instruments meeting genome-wide significance ($p < 5 \times 10^{-8}$) and applied clumping ($r^2 < 0.001$ within a 10,000 kb window) against the 1000 Genomes Phase 3 European reference. Variants were then harmonized across exposure and outcome datasets, excluding ambiguous palindromic SNPs. Multiple MR estimators were applied, including inverse variance weighted (IVW), MR Egger, weighted median, simple median, and weighted mode. Cochran's Q test was used to assess heterogeneity, and the MR-Egger intercept test for horizontal pleiotropy. Using MR-PRESSO to detect and correct horizontal pleiotropy by removing outliers[41]. As our analyses were confined to hypothesis-driven trait pairs, we report nominal $p < 0.05$ without further multiple-testing correction.

To further investigate the joint causal effects of T2D and proinsulin levels on DR, we extended the analysis to an MVMR framework. The IVW approach was utilized as the primary statistical method to evaluate the significance of variables within the MVMR analysis. Moreover, we used the ExPheWAS Browser's cis-MR module[42] to estimate the causal effect of an

exposure on an outcome by leveraging only genetic variation within the cis-region of a target gene based on the UKB cohort. For each gene, principal components capturing local genotype variation were computed and used as instruments in an inverse-variance weighted MR framework to test for a causal effect of a specified exposure phenotype on an outcome phenotype.

## Colocalization analysis

We conducted colocalization analysis using the coloc[43] package in R, which employs a Bayesian colocalization framework, to assess whether GWAS signals of DR and DR-related phenotypes, as well as significant tissue-specific eQTL signals, share the same causal variant. eQTL data were obtained from the GTEx Portal. We used the default prior probabilities ($p_1 = 1 \times 10^{-4}$, $p_2 = 1 \times 10^{-4}$, $p_{12} = 1 \times 10^{-5}$). The method computes posterior probabilities for five mutually exclusive hypotheses (H0–H4) based on approximate Bayes factors, which represent different scenarios regarding the presence and sharing of association signals between the two datasets. We particularly focused on the posterior probability of hypothesis 4 (PP.H4). A PP.H4 greater than 80% indicates strong evidence that the GWAS phenotype and eQTL signal likely share a common causal SNP, while PP.H3 indicates independent causal variants. Analyses were performed within a 500 kb flanking region around our UKB genome-wide significant SNP.

## Ethics approval

The research adhered to the tenets of the Declaration of Helsinki. Ethical approval for UK Biobank research was appropriately secured from the National Health Service National Research Ethics Service (reference 11/NW/0382). This study was approved by the Ethics Committee of the University of Nottingham, Ningbo, China. This study utilized replication cohorts from GoDARTS, FinnGen, and those from African American and European ancestries, which are incorporated into the U.S. Department of Veterans Affairs Million Veteran Program (MVP). For GoDARTS, Data provision and linkage were carried out by the University of Dundee Health Informatics Centre (HIC), with analysis of anonymized data performed in a Scottish Government-accredited secure safe haven. HIC Standard Operating Procedures have been reviewed and approved by the NHS East of Scotland Research Ethics Service. Ethical approval for this study was granted by the Tayside Medical Ethics Committee. For FinnGen, participants provided informed consent for biobank research under the Finnish Biobank Act, and the FinnGen study protocol was approved by the Coordinating Ethics Committee of the Hospital District of Helsinki and Uusimaa (HUS/990/2017) with recruitment following biobank protocols approved by Fimea. For MVP (AFR and EUR groups), all participants provided informed consent, and the study protocol was approved by the VA Central Institutional Review Board (protocol code MVP001, approved in 2010).

## Results

### GWAS results

We identified 16,988 samples, comprising 1824 cases and 15,164 controls. Supplementary Table 1 showed that age, sex, BMI, and DoD were all found to be significantly different ($p < 0.05$) between cases and controls. A total of 9,623,834 imputed SNPs passed from routine QC checking. Fig. 2 shows the Manhattan plot, Q-Q plot and regional plots. We identified one locus that reached genome-wide significance ($p < 5 \times 10^{-8}$), with a clear cluster of associated variants and a genomic inflation factor ($\lambda$) close to 1, indicating no appreciable inflation. The most significant SNP, rs6066146, is on chromosome 20 ($p = 4.21 \times 10^{-8}$) in *EYA2* (Table 1). The SNP-based heritability of DR was estimated to be 14.6% (standard deviation: 0.11) in this T2D population.

### Replication stage

rs6066146 replicated in the FinnGen, African American, and European samples with $p$ values of 0.008, 0.003, and 0.015, respectively (Table 2). We further assessed the replication of top DR-associated SNPs reported in five independent cohorts within our UKB DR GWAS. Three loci showed

nominal replication ($p < 0.05$): the African American ancestry SNPs rs7903146 ($p = 0.039$) and rs2237897 ($p = 0.037$), and the European ancestry SNP rs34872471 ($p = 0.049$). The Chinese GWAS[44] SNP rs1399634 was borderline significant ($p = 0.050$), while the remaining variants did not replicate (Supplementary Table 5).

## Gene, gene-set and tissue expression analysis by MAGMA

In the gene analysis, all the SNPs located within genes were mapped to 19,295 protein-coding genes. Genome-wide significance was defined (red dashed line in the plot) at $p = 0.05/19295 = 2.59 \times 10^{-6}$. No gene demonstrated the strongest association with DR, see Supplementary Fig. 1.

In the gene-set analysis, GOBP_REGULATION_OF_PRO_B_CELL_DIFFERENTIATION was the most significant set, achieving Bonferroni-corrected significance with a $p$ value of $1.98 \times 10^{-6}$ (<0.05/10,678 = $4.68 \times 10^{-6}$). The top 30 gene sets are shown in Fig. 3A and Supplementary Table 6. Within this gene set, we evaluated individual genes to pinpoint contributors to the observed significance. Notably, as shown in Supplementary Table 7 and Fig. 3B, *SOS1* and *SOS2* exhibit notable *Z*-scores ($p < 0.05$), indicating that they may play key roles in driving the signal.

In the tissue expression analysis, the spleen demonstrated statistical significance in the expression analysis of 54 specific tissue types but not in 30 general tissue types, indicating a significant association with DR (Fig. 3C, D and Supplementary Table 8).

## Cis-eQTL, chromatin interaction analysis, and hypergeometric tests

Seventeen annotated SNPs from ANNOVAR were used for cis-eQTL mapping, and seven SNPs were identified as significantly associated with the expression of the *EYA2* gene in thyroid tissue, with an FDR < 0.05. Chromatin interaction analysis revealed that 104 interaction regions are significant, with the occurring tissues or cells being Mesenchymal Stem Cells, Mesendoderm, IMR90, Trophoblast-like Cells, hESCs, Liver, and Spleen (Supplementary Table 9). The gene mapping of chromatin interactions and cis-eQTL is presented in Fig. 4. Genes interacting with *EYA2* were *NCOA3*, *ZMYND8*, *SLC2A10*, and *ZNF334*. The hypergeometric tests revealed a significant overrepresentation of our genes in GWAS Catalog gene sets associated with waist-to-hip ratio adjusted for BMI ($p < 0.05$ and Supplementary Fig. 2).

## Evaluation of ocular tissue expression and phenome-wide association study (PheWAS)

In the Single Cell Type analysis of the eye, *EYA2* exhibited the most prominent expression in cone photoreceptor cells (8.5 nTPM), followed by rod photoreceptor cells (2.2 nTPM) and horizontal cells (1.3 nTPM). Tabula Sapiens data further demonstrated strong expression levels of *EYA2* in CD8[+] alpha-beta T cells (1813.7 nTPM) (Supplementary Fig. 3). FANTOM data indicated elevated *EYA2* expression in the retina (62.5 scaled tags per million).

The PheWAS of rs6066146 highlighted its top 10 significant associations across three domains (Supplementary Fig. 4 and Supplementary Table 10): ocular traits (intraocular pressure [IOP], $p = 1.48 \times 10^{-3}$), diabetic outcomes (T2D, $p = 1.78 \times 10^{-3}$; DR, $p = 2.32 \times 10^{-3}$), and metabolic measures (triglycerides, $p = 1.23 \times 10^{-4}$; HDL cholesterol, $p = 7.94 \times 10^{-4}$), underscoring this variant's pleiotropic influence on both systemic metabolism and ocular physiology.

## TWAS results

TWAS for *EYA2* was restricted to the spleen and thyroid based on prior screens. Variant rs6066146 exhibited tissue-specific effects on *EYA2* expression, showing a significant association in thyroid (TWAS.Z = −2.10, TWAS.P = 0.035) and in spleen (TWAS.Z = 2.19, TWAS.P = 0.028; Supplementary Table 11). The estimated heritability of *EYA2* expression was approximately 0.30 in both the thyroid and the spleen.

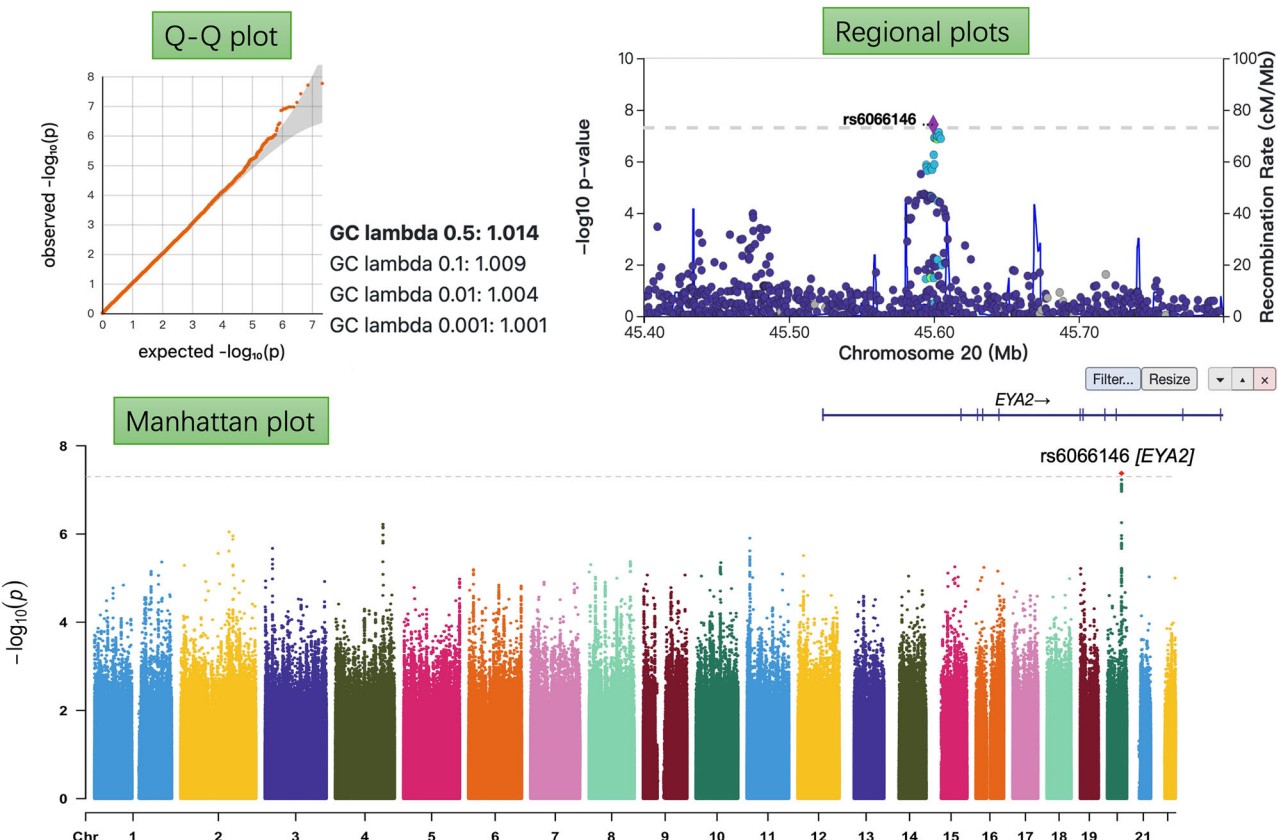

**Fig. 2 | The Manhattan plot, regional plots and Q–Q plot of the GWAS analysis on diabetic retinopathy from type 2 diabetes ($N = 16,988$).** Single-variant association testing was performed using a fastGWA-GLMM generalized linear mixed model implemented in GCTA, and two-sided $p$ values were obtained from a 1 d.f. score test of the SNP effect. The dashed gray line in the Manhattan plot indicates the genome-wide significance threshold ($p = 5 \times 10^{-8}$) used to account for multiple comparisons. The regional plot highlights a cluster of association signals in the gene *EYA2*. The grey-shaded area in the Q–Q plot denotes the 95% confidence band expected under the null hypothesis, and the genomic inflation factor (λGC) is close to 1, indicating no significant population stratification.

**Table 1 | Significant and suggestive loci from GWAS of diabetic retinopathy using UKB**

| Marker | Chr | Position | Nearest gene | EA | NEA | AF1 | Beta | SE | *P* values |
|---|---|---|---|---|---|---|---|---|---|
| rs6066146 | 20 | 45599553 | *EYA2* | A | G | 0.51 | 0.02 | 0.003 | $4.21 \times 10^{-8a}$ |
| rs13041611 | 20 | 45600309 | *EYA2* | C | T | 0.53 | −0.02 | 0.003 | $7.75 \times 10^{-8}$ |
| rs13043230 | 20 | 45600617 | *EYA2* | T | C | 0.53 | −0.02 | 0.003 | $7.78 \times 10^{-8}$ |
| rs11697925 | 20 | 45602129 | *EYA2* | A | G | 0.53 | −0.02 | 0.003 | $8.99 \times 10^{-8}$ |
| rs13042954 | 20 | 45602193 | *EYA2* | G | A | 0.53 | −0.02 | 0.003 | $9.74 \times 10^{-8}$ |
| rs13042847 | 20 | 45602299 | *EYA2* | C | T | 0.53 | −0.02 | 0.003 | $7.50 \times 10^{-8}$ |
| rs13043269 | 20 | 45602416 | *EYA2* | G | A | 0.53 | −0.02 | 0.003 | $7.33 \times 10^{-8}$ |
| rs13039645 | 20 | 45603778 | *EYA2* | G | A | 0.53 | −0.02 | 0.003 | $8.18 \times 10^{-8}$ |

Association testing was performed using a generalized linear mixed model. Two-sided $p$ values were obtained from the 1 d.f. score test of the SNP effect. Multiple comparisons were addressed, and the conventional genome-wide significance threshold is $p = 5 \times 10^{-8}$.
*Chr* chromosome, *EA* effect allele, *NEA* non-effect allele, *AF1* frequency of effect allele.
[a]indicates SNPs reach genome-wide significance with $p$ value $< 5 \times 10^{-8}$, while others are suggestive.

## Mendelian randomization results

Overall, when each trait was treated as the exposure and DR as the outcome, significant causal effects were detected in forward-direction MR but not in reverse-direction analyses. Specifically, T2D showed significant effects in 23 of 41 tests (IVW $p < 0.05$), proinsulin levels in 1 of 1 test, eye or eyelid problems in 1 of 5 tests, and glaucoma in 1 of 2 tests. Cataract showed no significant association in either direction. Notably, T2D adjusted for BMI (IEU GWAS ID: ebi-a-GCST007516), proinsulin (ebi-a-GCST001212), eye or eyelid problem (ebi-a-GCST90038640), and glaucoma (ukb-b-17324) all exhibited significant associations with DR. Neither heterogeneity nor horizontal pleiotropy reached statistical significance ($p > 0.05$; Supplementary Table 12). SNP effect estimates for these traits versus DR are shown in Fig. 5, and the corresponding sensitivity analyses plots are in Supplementary Fig. 5. A subsequent cis-MR at the *EYA2* locus, where the gene-T2D association in ExPheWas is $p = 3.0 \times 10^{-7}$, estimated an IVW odds ratio of 2.52 (95% CI: 1.22–5.21) for T2D on DR ($p = 0.013$), providing robust, gene-centric evidence of a T2D-DR causal pathway (Supplementary Fig. 6).

In the MVMR analysis, we harmonized three independent SNPs associated with BMI-adjusted T2D (ebi-a-GCST007516) and proinsulin

## Table 2 | Replication of lead SNPs identified in the UK Biobank (UKB) GWAS in independent cohorts

| | Cohorts | SNP | Chr | Pos | EA | NEA | AF1 | Beta(SE) | P | Distance |
|---|---|---|---|---|---|---|---|---|---|---|
| Discovery cohort | UK Biobank | rs6066146 | 20 | 45599553 | A | G | 0.51 | 0.020(0.003) | $4.21 \times 10^{-8}$ | 0 |
| Replication cohorts | GoDARTS[23] | rs6066146 | 20 | 45599553 | A | G | 0.51 | 0.024(0.036) | $5.07 \times 10^{-1}$ | 0 |
| | FinnGen[24] | rs6066146 | 20 | 45599553 | A | G | 0.56 | 0.067(0.025) | $8.00 \times 10^{-3}$ | 0 |
| | African American[25] | rs6066146 | 20 | 45599553 | A | G | 0.71 | 0.050(0.017) | $3.00 \times 10^{-3}$ | 0 |
| | European[25] | rs6066146 | 20 | 45599553 | A | G | 0.53 | 0.017(0.010) | $1.50 \times 10^{-2}$ | 0 |

Association results (Beta, SE, and $p$) for the lead SNPs were obtained from UKB GWAS summary statistics. UKB discovery associations were tested using a generalized linear mixed model, with two-sided $p$ values derived from a 1 d.f. score test of the SNP effect. For replication cohorts, $p$ values shown are as reported in the corresponding GWAS summary statistics. *Chr* chromosome, *Pos* position, *EA* effect allele, *NEA* non-effect allele, *AF1* frequency of effect allele. The effect alleles and Beta values of the SNPs from other cohorts were adjusted to match the strand direction of the UKB, ensuring consistency across the datasets.

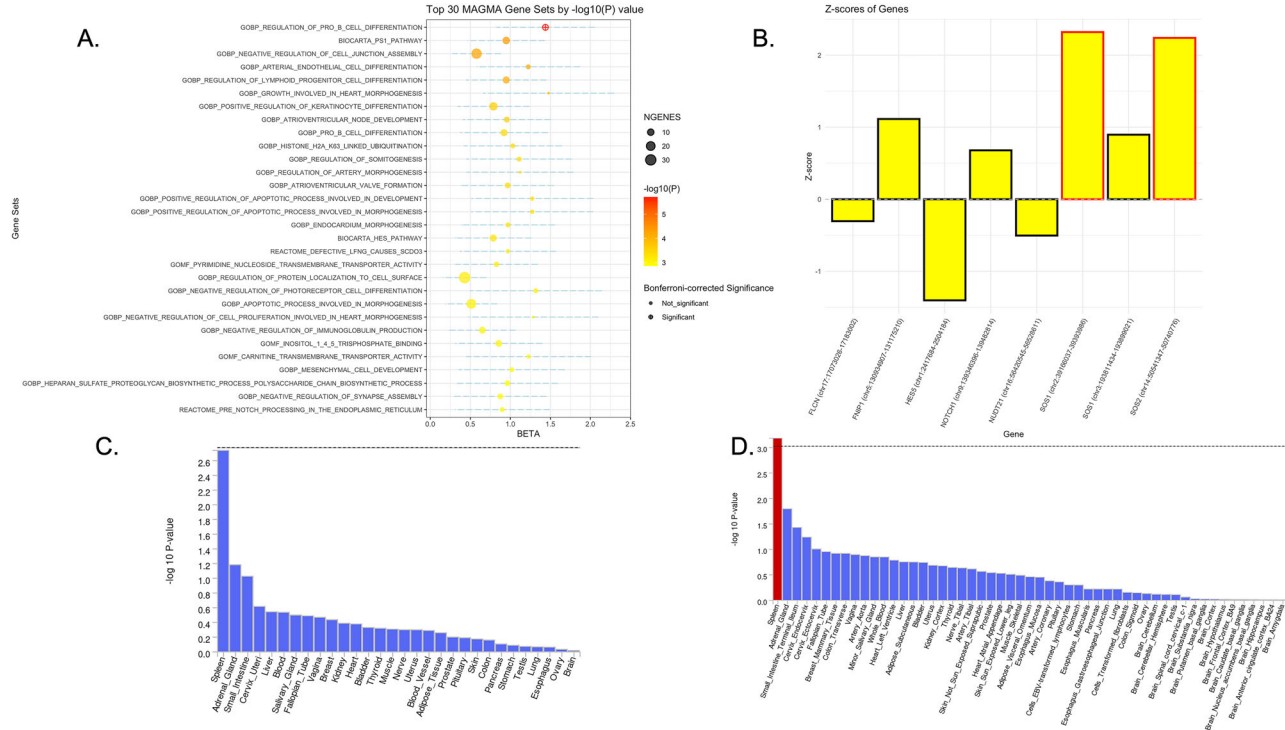

**Fig. 3 | Gene-set and tissue expression analysis results. A** Top 30 gene set analysis results by MAGMA. Gene-set enrichment was tested using MAGMA competitive gene-set analysis, which fits a gene-level regression model of gene association statistics on gene-set membership. *P* values correspond to a one-sided test of the gene-set effect. In the figure, each bubble represents a MAGMA gene set. The vertical axis displays the top 30 gene sets ranked by $-\log_{10}(p)$, while the horizontal axis shows their effect size (BETA). Bubble size corresponds to the number of genes in each set, and bubble color indicates the level of $-\log_{10}(p)$. Bonferroni-corrected significant sets are marked with an encircled plus sign. **B** Statistical details of genes within GOBP_REGULATION_OF_PRO_B_CELL_DIFFERENTIATION. Each bar represents a gene's Z-score, with taller bars (above zero) indicating a positive association and lower bars (below zero) showing a negative association. Genes *SOS1* and *SOS2* with significant signals are highlighted in red boxes. **C, D** Tissue expression results on 30 general tissue types and 54 specific tissue types by GTEx v7 in the FUMA. The dashed line shows the cut-off $p$ value for significance with Bonferroni adjustment for multiple hypothesis testing.

levels (ebi-a-GCST001212) as instruments for simultaneous exposures. Genetically predicted proinsulin levels remained significantly associated with DR ($p = 0.03$), whereas genetically predicted T2D risk did not ($p = 0.17$; Supplementary Table 13). Further analyses indicated a significant directional effect from BMI-adjusted T2D to proinsulin, but not vice versa (Supplementary Table 14). These findings support a mediating role of proinsulin in the causal pathway from BMI-adjusted T2D to DR.

### Colocalization analysis results
Using T2D summary statistics from Mahajan et al.[45] and our UKB DR GWAS at the *EYA2* locus, colocalization yielded PP.H3 = 0.433 and PP.H4 = 0.553, reflecting moderate support for a shared causal variant alongside independent effects, though PP.H4 falls below the conventional

>80% threshold for definitive colocalization (Supplementary Table 15 and Supplementary Fig. 7). Neither thyroid and spleen eQTL signals for *EYA2*, nor the GWAS signals for proinsulin, colocalized with the DR association peak (PP.H4 < 0.80).

### Discussion
In our UKB DR GWAS, the *EYA2* variant rs6066146 not only reached genome-wide significance ($p < 5 \times 10^{-8}$) with minimal inflation ($\lambda = 1.003$; Fig. 2) but also replicated in FinnGen, African American, and European ancestry cohorts. Notably, *EYA2* has been identified as an active promoter for T2D[45] and also emerges as a key driver of DR. Our colocalization analysis at the *EYA2* locus provided suggestive evidence of a shared causal variant (PP.H4 = 0.553) alongside independent association signals (PP.H3 = 0.433),

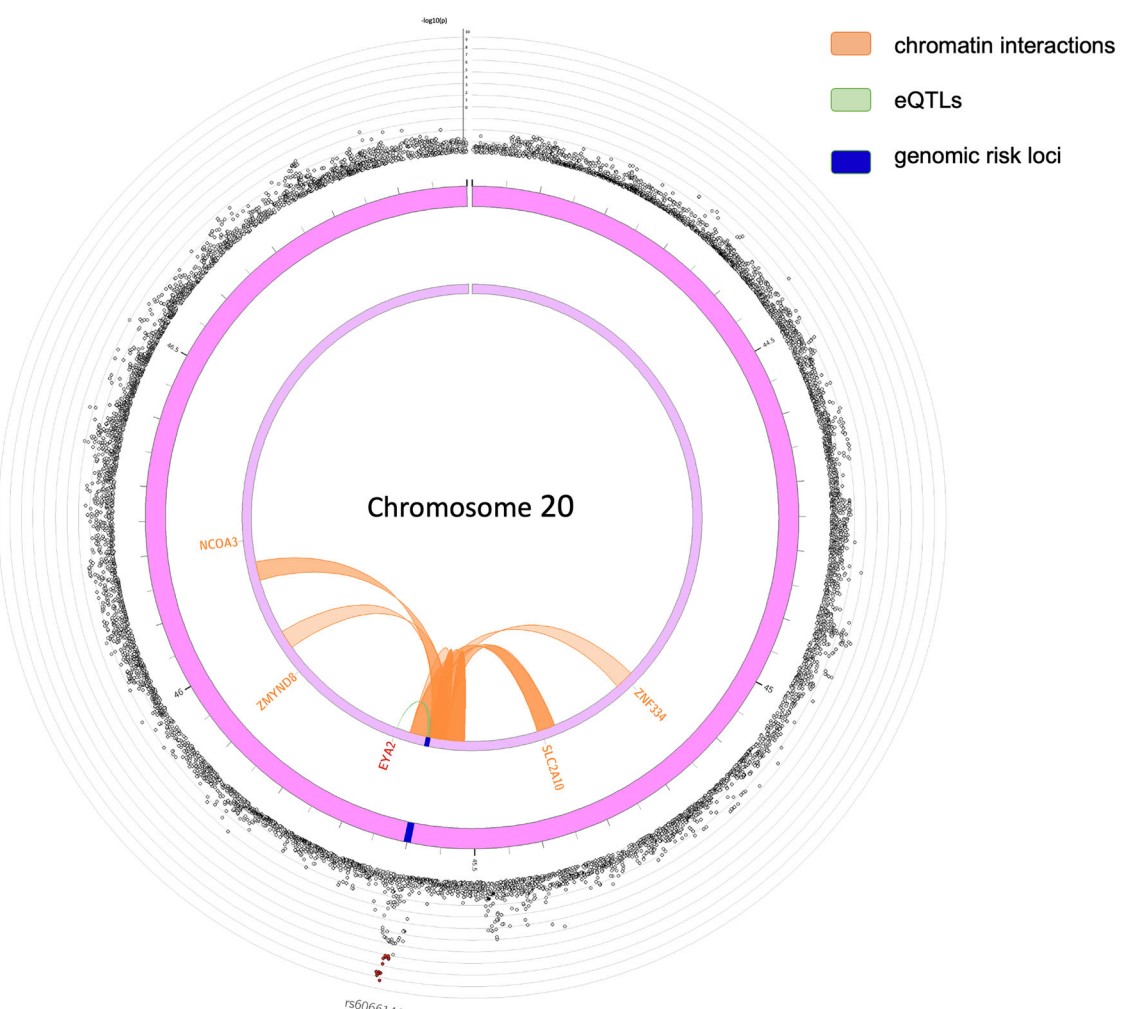

**Fig. 4 | Circus plot of gene mapping of chromatin interactions and cis-eQTL.** The top SNPs in each risk locus are displayed in the outermost layer (Manhattan plot). The second and third layers are chromosome rings, where genomic risk loci are highlighted in blue to indicate their locations. Genes mapped exclusively by chromatin interactions are colored orange, exclusively by eQTLs are colored green, and those mapped by both are colored red. Gene *EYA2* is colored red.

and subsequent cis-MR provided further evidence that genetically driven T2D variation at this gene exerts a significant causal effect on DR.

The *EYA2* gene, which stands for "Eyes Absent Homolog 2", encodes a member of the eyes absent family of proteins, which encodes proteins that may be post-translationally modified and may play a role in eye development[46]. Zhang et al.[47] discovered that *EYA2* is expressed in previously unreported sensory and developmental systems, including the eye, with evidence of β-Gal activity in various eye structures during retinal development. Previous studies suggested that *EYA2* may be involved in retinal neuronal differentiation. The role of human *EYA* paralogs (*EYA1-4*) was first identified in *Drosophila* eye development, specifically within the retinal determination gene network[48]. Further research showed that inhibition of *EYA* protein tyrosine phosphatase activity slows vascular growth in the postnatal *mouse* retina[49]. An experiment using a mouse model of oxygen-induced retinopathy showed that *EYA* inhibitors reduced pathological neovascularization in a proliferative retinopathy model[50]. The tyrosine phosphatase activity of *EYA* is critical to the choice between repair and apoptosis following DNA damage and may promote angiogenesis under hypoxia by dephosphorylating *H2AX*[46]. In summary, evidence from several studies supported the pro-angiogenic role of *EYA* tyrosine phosphatase activity[46,49,51–53]. Tadjuidje et al. demonstrate that *EYA* is expressed in endothelial cells and its tyrosine phosphatase activity promotes angiogenesis[51].

The gene set GOBP_REGULATION_OF_PRO_B_CELL_DIFFER-ENTIATION reached Bonferroni-corrected significance in our MAGMA analysis, with *SOS1* and *SOS2* driving this signal. The spleen emerged as a significant tissue in both our tissue expression and chromatin-interaction analyses (the loci at chr20:45,560,001–45,600,000 and 45,600,001–45,640,000 showed strong interactions) and received nominal support from TWAS. These findings align with murine studies demonstrating the relationship between the spleen and B cells. *SOS1/2* double-knockout mice exhibit a significant reduction of mature B cells in the spleen[54]. *GRB2*, which is highly associated with DR in T2D patients[21], is likewise essential for pro-B cell survival and maturation, with its absence depleting newly mature B cells in the spleen[55]. Overall, splenic B-cell abundance could serve as a predictive biomarker for DR risk.

Cis-eQTL analysis in thyroid tissue demonstrated that the allelic configuration of rs6066146 significantly modulates *EYA2* expression, and thyroid tissue also showed nominal significance in our TWAS. Although we did not uncover direct pathway evidence linking thyroid to DR in our study, several studies support a clinical connection. Subclinical hypothyroidism is associated with an increased risk of sight-threatening DR[56], and in euthyroid T2D patients, free triiodothyronine levels within the normal range are negatively associated with DR[57]. Moreover, thyroid dysfunction alters *VEGF* levels, a key driver of retinal angiogenesis and permeability, and elevated *VEGF* in Graves' disease and Hashimoto's thyroiditis may exacerbate DR progression[58].

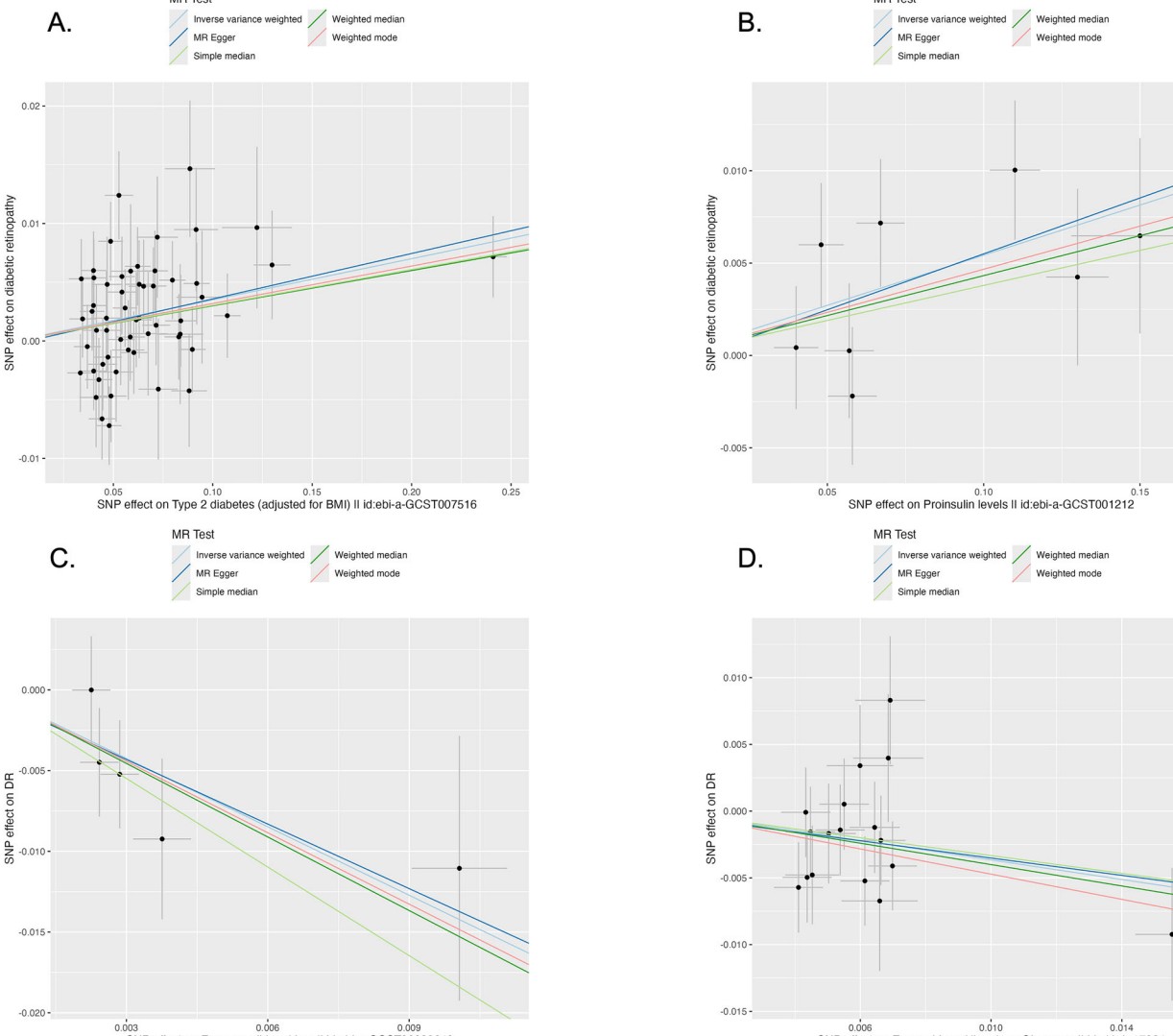

**Fig. 5 | Mendelian randomization scatter plots for diabetic retinopathy (DR).** SNP-exposure effect estimates (x-axis) are plotted against SNP-DR effect estimates (y-axis). Each black point represents one LD-independent instrumental SNP (n = number of instruments): **A** T2D (adjusted for BMI), n = 56. **B** Proinsulin, n = 8. **C** Eye or Eyelid Problems, n = 5. **D** Glaucoma, n = 17. Each black dot denotes the SNP effect point estimate (BETA); the horizontal and vertical error bars are centered on BETA and indicate the 95% confidence intervals for the SNP-exposure and SNP-DR associations, respectively. The colored lines represent Mendelian Randomization (MR) estimates from inverse variance weighted, MR Egger, simple median, weighted median, and weighted mode methods.

Our exploratory MVMR analyses identified proinsulin as a significant mediator of T2D on DR, suggesting its contribution to DR risk, which may account for the moderate evidence for colocalization between DR and T2D. Locus-level colocalization around key GWAS peaks indicates that no single variant drives both proinsulin variation and DR susceptibility. This pattern suggests that proinsulin's mediating effect arises from the combined action of many small-effect alleles rather than one shared causal SNP. Notably, C-peptide from proinsulin proteolysis prevents blood-retinal barrier disruption and pathological neovascularization through modulation, making it a promising agent for preventing and treating DR[36]. In addition, in univariable MR analyses, using eyelid disorders and glaucoma as exposures and DR as the outcome, revealed significant associations that implicate potential ocular vascular and inflammatory pathways in DR pathogenesis. Indeed, diabetic patients without PDR exhibit a higher percentage of Meibomian gland (MG) dropout and elevated meiboscore grades than non-diabetic individuals, reflecting more severe MG dysfunction[38]. Moreover, MG loss scores vary significantly across DR severity groups, further linking DR to eyelid abnormalities[59]. Similarly, T2D patients with primary open-angle glaucoma have a threefold increased risk of developing DR compared to

those without glaucoma[60]. However, the causal relationship between eyelid problems, glaucoma, and DR remains underexplored, with limited studies addressing causality.

Hypergeometric analysis revealed a significant enrichment of WHR-associated gene sets among our DR-mapped genes (*EYA2*, *ZNF334*, *SLC2A10*, and *ZMYND8*), consistent with clinical evidence that a low waist-to-hip ratio protects against DR in T2D, while a higher WHR is linked to both its presence and severity[61,62]. PheWAS analysis linked our lead SNP to IOP, echoing earlier reports of significantly elevated IOP in diabetic patients and its positive correlation with HbA1c in those with DR[63]. Moreover, a recent MR analysis confirmed that higher IOP is causally associated with, and a risk factor for DR[64].

We acknowledge that the lack of stratification by DR severity in the UKB is a limitation. Another objective limitation of our study is the accuracy of DR identification in the UKB, which utilizes hospital episode statistics linkage. Compared to the GWAS of DR conducted by Forrest et al. using the UKB cohort[65], which did not identify genome-wide significant associations, our study exclusively analyzed white British DR cases defined by the ICD-10 code "H360" (1824 cases) instead of those identified by ICD-10 code "E11.3"

or through self-reporting (866 cases). By incorporating a larger number of DR cases and including key covariates such as duration of T2D levels, our GWAS approach enhances the potential power to identify more risk loci. Since our GWAS analysis included DR cases with comorbid AMD and other retinal diseases, we acknowledge that we sacrificed some phenotypic purity but gained more case samples, thereby increasing statistical power. In sensitivity analyses excluding 270 AMD cases (≈15% of the original 1824 DR cases), the *EYA2* variant rs6066146 remained suggestively associated with DR ($p = 5.8 \times 10^{-6}$; Supplementary Fig. 8) with BETA(SE) = 0.015(0.003). The slight attenuation of significance likely reflects the high co-occurrence of AMD and DR in older individuals (Supplementary Fig. 9), as removing AMD cases also disproportionately removes older DR cases, thereby reducing power. Crucially, the persistence and consistent direction of the DR-association demonstrate that residual AMD comorbidity in our T2D-restricted sample does not materially affect the validity of the *EYA2*-DR signal.

Another weakness of our study is in the replication stage. We used cohorts such as FinnGen and, similarly, African American and European, which contain population controls rather than diabetes-specific controls in their GWAS on DR. This study design may elevate type I error, since true signals can fail to replicate in less well-defined replication cohorts. Fortunately, our downstream analyses demonstrate that the rs6066146 variant influences both T2D and DR, and that, within the *EYA2* region, T2D predisposes to DR. However, it is still recommended that the results generated should be interpreted with caution. For the GoDARTS study, although using the diabetes controls (Supplementary Table 2), it also included a subset of T1D cases, which reduced power to replicate our T2D-specific DR signal. Despite these limitations, the consistent replication ($p < 0.05$ & BETA > 0) of rs6066146 across three cohorts supports its robustness across ancestries. Moreover, when we examined top DR-associated SNPs from other discovery cohorts in our GWAS summary statistics, three showed nominal replication, indicating that differences in phenotype definitions, ancestral heterogeneity, and variable case-ascertainment protocols did not detract from the substantive insights (Supplementary Table 5). Since our study is limited to the white British population and most downstream analyses used European LD panels, additional studies in non-European populations are still required to confirm the generalizability of the *EYA2* association.

## Conclusion
Our study demonstrated that the *EYA2* gene had associations with DR in T2D patients and, through a multifaceted approach, illuminated the complex pathophysiology of this disease. By deepening our mechanistic understanding of DR, this work offers avenues for future research.

## Data availability
The individual-level genotype and phenotype data used in this study are available from the UK Biobank under controlled access and can be obtained by approved researchers through application to the UK Biobank. The source data underlying the figures presented in this study are provided in the Supplementary Information. Specifically, the summary statistics data for the Manhattan plot in Fig. 2 is provided in Supplementary Data. The source data for Fig. 3A are in Supplementary Table 6. The source data for Fig. 3B are in Supplementary Table 7. The source data for Fig. 3C, D are in Supplementary Table 8. The source data for Fig. 4 are in Supplementary Table 9. The source data for Fig. 5 are in Supplementary Table 12. GoDARTS summary statistics used in this study were provided by Aravind L. Rajendrakumar from his PhD thesis and will be made available to qualified researchers upon reasonable request. Other summary statistics for Mendelian randomization are available in the IEU OpenGWAS database. The eQTL data used for the analyses described in this manuscript were obtained from the GTEx Portal on 05/10/2025. Should any data pertinent to this study need to be presented within this paper or its other files, the authors can provide such data upon reasonable request.

## Code availability
Custom scripts used for data processing and analysis are available from the corresponding authors upon reasonable request.

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

## Acknowledgements

The authors gratefully thank all the participants and professionals contributing to the UKB, GoDARTS, FinnGen, and African American &

European ancestry studies. This study adheres to all ethical guidelines and data protection protocols of the UK Biobank. The current study was conducted under the approved UK Biobank data application number 50604. This study was mainly funded by the Pioneer and Leading Goose R&D Program of Zhejiang Province 2023, with reference number 2023C04049, and the Ningbo International Collaboration Program 2023, with reference number 2023H025.

## Author contributions

T.C. drafted the paper and performed the GWAS analysis. Q.P. and Y.T. contributed to data formatting and correction. C.N., A.R., Y.Y., T.D., M.H., and C.P. provided comments on the paper. Y.S. and W.M. organized the project and provided comments.

## Competing interests

The authors declare no competing interests.
