## [Transparent Peer Review File · Communications Medicine]

A genome-wide association study identifies *EYA2* as a contributing gene for diabetic retinopathy in type 2 diabetes

Corresponding Author: Dr Weihua Meng

Version 0:

Reviewer comments:

Reviewer #1

(Remarks to the Author)

This study aims to identify genetic variants associated with diabetic retinopathy (DR) in patients with type 2 diabetes using data from the UK Biobank. A genome-wide association study (GWAS) identified a significant single nucleotide polymorphism (SNP), rs6066146, in the *EYA2* gene. This association was confirmed through replication and meta-analysis with additional cohorts from GoDARTS and FinnGen. The SNP's impact on gene expression was analyzed through tissue expression and cis-eQTL analyses, showing significant expression in thyroid and spleen tissues. Moreover, Mendelian Randomization analysis suggested connections between DR, eyelid problems, and glaucoma. The study concludes that *EYA2* is a genomic risk locus for DR. Overall, this study presents a GWAS on DR using a refined definition of cases and controls from the UK Biobank, identifying *EYA2* as a potentially new biologically relevant gene for DR risk. The manuscript is well-written. However, there are critical methodological issues that require clarification to fully substantiate the findings. These concerns must be addressed to enhance the credibility and impact of the study.

1. Although a refined case-control population from the UK Biobank was initially tested, the discovery population consists of subjects without diabetes mellitus and the addition of FinnGen dataset in the subsequent analysis further introduced healthy subjects into the analysis. The use of population controls for comparison with DR cases could introduce biases and should be thoroughly discussed.
2. Please define the specific flanking regions included in the gene set test to clarify the scope of the analysis.
3. It was unclear how many SNPs were initially screened to identify the seven cis-eQTL SNPs reported. Detailing this process would enhance the transparency of the methods.
4. In the transcriptome-wide association study (TWAS):
 - Clarify the relevance of thyroid and spleen tissues to DR.
 - Specify the total number of tissues tested and whether Bonferroni correction was applied for multiple testing.
5. Regarding the Mendelian Randomization (MR) analysis:
 - State the total number of ocular traits examined.
 - Justify the selection of "Eye or Eyelid Problems" and glaucoma as exposures for the analysis.
6. Discuss whether any known DR-associated loci were replicated in this GWAS or if they appeared at least nominally significant, providing more detailed insights into the consistency of these findings with previous studies.
7. The inclusion of the FinnGen cohort might introduce bias due to its use of population controls. Please discuss the implications of this for the genetic association results.
8. Given that variants in *EYA2* have been previously associated with type 2 diabetes (T2D) in other studies (Nat Genet. 2018;50(11):1505–1513. doi: 10.1038/s41588-018-0241-6), highlight the evidence supporting the specificity of *EYA2*'s involvement in DR, distinguishing it from general T2D susceptibility.

Reviewer #2

(Remarks to the Author)

The authors conducted a GWAS using the UK Biobank and identified a genome-wide significant association between diabetic retinopathy (DR) and the *EYA2* gene variant rs6066146 in type 2 diabetes patients. The association was replicated in the FinnGen cohort, and functional analyses suggest involvement of immune and angiogenic pathways. Mendelian randomization further identified glaucoma and eyelid problems as potential risk factors for DR. Please see the specific

comments below for details.

1. While DR cases were identified using ICD-10 code H360, it is unclear whether individuals with other retinal diseases (e.g., age-related macular degeneration) were excluded. Please clarify whether such conditions were considered and how misclassification risk was managed.
2. A more detailed comparison of the discovery and replication cohorts—particularly regarding phenotype definitions and disease ascertainment—would help readers better understand the replication results.
3. Please provide more detail on key parameters used in TWAS and Mendelian randomization analyses, such as SNP clumping thresholds, eQTL reference panels, and multiple testing correction approaches.
4. Since the GWAS was limited to a white British population, a brief discussion of how findings may or may not generalize to other ancestries would be valuable.
5. The manuscript reports spleen tissue enrichment and a B cell–related gene set, but the connection between the two is not discussed. Given the spleen’s role in B cell biology, a brief interpretation of this potential link may strengthen the discussion.

Version 1:

Reviewer comments:

Reviewer #1

(Remarks to the Author)

Thank you for your responses. The authors have successfully addressed most of my concerns.

The images or other third party material in this Peer Review File are included in the article’s Creative Commons license, unless indicated otherwise in a credit line to the material. If material is not included in the article’s Creative Commons license and your intended use is not permitted by statutory regulation or exceeds the permitted use, you will need to obtain permission directly from the copyright holder.

Dear reviewers:

Thank you for your valuable and insightful feedback on our manuscript. We greatly appreciate the time and effort you have taken to review our work and provide constructive comments. Your suggestions have been instrumental in guiding us to improve the quality and rigor of our submission. We have carefully considered your points and have addressed each of them in detail below. All page numbers refer to the clean manuscript file.

Reviewer #1 (Remarks to the Author):

This study aims to identify genetic variants associated with diabetic retinopathy (DR) in patients with type 2 diabetes using data from the UK Biobank. A genome-wide association study (GWAS) identified a significant single-nucleotide polymorphism (SNP), rs6066146, in the EYA2 gene. This association was confirmed through replication and meta-analysis with additional cohorts from GoDARTS and FinnGen. The SNP's impact on gene expression was analyzed through tissue expression and cis-eQTL analyses, showing significant expression in thyroid and spleen tissues. Moreover, Mendelian Randomization analysis suggested connections between DR, eyelid problems, and glaucoma. The study concludes that EYA2 is a genomic risk locus for DR. Overall, this study presents a GWAS on DR using a refined definition of cases and controls from the UK Biobank, identifying EYA2 as a potentially new biologically relevant gene for DR risk. The manuscript is well-written. However, there are critical methodological issues that require clarification to fully substantiate the findings. These concerns must be addressed to enhance the credibility and impact of the study.

1. Although a refined case-control population from the UK Biobank was initially tested, the discovery population consists of subjects without diabetes mellitus and the addition of FinnGen dataset in the subsequent analysis further introduced healthy subjects into the analysis. The use of population controls for comparison with DR cases could introduce biases and should be thoroughly discussed.

Response: We appreciate the reviewer's concern about potential bias arising from non-diabetic controls. In fact, our UK Biobank discovery GWAS was restricted to individuals with diabetes mellitus (both cases and controls). Specifically, all diabetic participants were defined by UK Biobank field ID 2443 ("diabetes diagnosed by doctors"). Thus, no non-diabetic subjects were

included in our primary UKB discovery. We apologize for any lack of clarity in the original manuscript and have added an explicit statement in lines 50-55 of the revised Methods section.

Methods: “We defined cases and controls by phenotype filtering using the following UKB field ID codes (Fig. 1). Only White British participants (field ID: 21000) were included. All samples were drawn from individuals with T2D, identified by a doctor’s diagnosis (field ID: 2443) and a “No” response to initiating insulin within one year (field ID: 2986). Among T2D patients, DR cases were those with ICD-10 code “H360” (field ID: 41202), while controls showed no record of this code.”

We acknowledge that using population controls in FinnGen, and likewise in the African American and European replication cohorts, can introduce bias by mixing up diabetes-related and DR-specific signals, especially when acting as a discovery cohort. The degree of bias depends on whether significant signals drive DR only, diabetes only, or both DR and diabetes. We think the impact or biases will be more severe if FinnGen is used as a discovery cohort in a GWAS, as it will introduce more type I and type II errors. In our study, however, FinnGen functions solely as an independent replication cohort. We first identified DR signals in a discovery cohort in which both cases and controls were all T2D, and we then sought to replicate these signals in FinnGen, which employed population controls. Consequently, the principal concern of using a population control in the replication stage is an elevated type I error, where spurious associations may be introduced. We discussed it in lines 376-381.

BTW: In a GWAS in which population controls are used in a discovery stage, population controls can enhance sensitivity for T2D loci and shared T2D-DR loci, while introducing biases for DR-specific signals, which therefore warrant replication in cohorts without diabetic confounding. But it is not the case for our study, as we only use FinnGen as a replication.

Discussion: “Another weakness of our study is in the replication stage. We used cohorts such as FinnGen and similarly the African American and European, which contain population controls rather than diabetes-specific controls in their GWAS on DR. This study design may elevate type I error, since true signals can fail to replicate in less well-defined replication cohorts. Fortunately, our downstream analyses demonstrate that the rs6066146 variant influences both T2D and DR, and that, within the EYA2 region, T2D predisposes to DR. However, it is still recommended that the results generated should be interpreted with caution.”

2. Please define the specific flanking regions included in the gene set test to clarify the scope of the analysis.

Response: Thank you for your valuable suggestion. We applied FUMA's default naive positional mapping in MAGMA v1.08, which assigns SNPs to genes if they fall within 10 kb upstream or downstream of the annotated gene body. Gene-level p-values are then computed and tested across 10,678 MSigDB v6.2 gene sets (4,761 curated sets + 5,917 GO terms) with a strict Bonferroni correction. We have added the definition of these flanking regions in Methods (lines 95-98).

Methods: *“(ii) to conduct gene-set analyses by mapping SNPs to genes using default naive positional mapping-assigning SNPs within 10 kb upstream or downstream of the annotated gene body. Gene-level p-values were then computed and tested across 10,678 MSigDB v6.2 gene sets (4,761 curated sets + 5,917 GO terms) with Bonferroni correction.”*

3. It was unclear how many SNPs were initially screened to identify the seven cis-eQTL SNPs reported. Detailing this process would enhance the transparency of the methods.

Response: We appreciate your insightful comment. FUMA first identified a single lead SNP (rs6066146) from our DR GWAS and then, using ANNOVAR annotation, retrieved 17 intronic variants in high LD with that lead SNP. These 17 SNPs were submitted to GTEx v7 and v8 cis-eQTL mapping (± 1 Mb window), and seven of them passed the $FDR < 0.05$ threshold. We have added the details in the Methods (lines 102-106) and Results (lines 229-231) sections.

Methods: *“We annotated all lead and proxy SNPs with ANNOVAR within FUMA. Positional mapping then assigned each SNP to any gene whose coding region, including a 10 kb upstream and downstream flanking window, overlapped the SNP's genomic coordinate. Using this ANNOVAR-derived SNP list, we next carried out cis-eQTL mapping with GTEx v7 and v8, testing every SNP within ± 1 Mb of a gene for expression association and retaining SNP-gene pairs that met a False Discovery Rate ($FDR < 0.05$).”*

Results: *“Seventeen annotated SNPs from ANNOVAR were used for cis-eQTL mapping, and seven SNPs were identified as significantly associated with the expression of the EYA2 gene in thyroid tissue, with an $FDR < 0.05$.”*

4. In the transcriptome-wide association study (TWAS):

- Clarify the relevance of thyroid and spleen tissues to DR.
- Specify the total number of tissues tested and whether Bonferroni correction was applied for multiple testing.

Response: Thank you for this valuable suggestion. We have clarified the relevance of thyroid

and spleen tissues to DR in the Discussion section (lines 322-338) by integrating our downstream analyses and literature findings.

Discussion: “The gene set GOBP_REGULATION_OF_PRO_B_CELL_DIFFERENTIATION reached genome-wide significance in our MAGMA analysis, with SOS1 and SOS2 driving this signal. Consistent with this, the spleen emerged as a significant tissue in both our tissue expression and chromatin-interaction analyses (the loci at chr20:45,560,001-45,600,000 and 45,600,001-45,640,000 showed strong interactions) and received nominal support from TWAS. These findings align with murine studies demonstrating the relationship between the spleen and B cells. SOS1/2 double-knockout mice exhibit a significant reduction of mature B cells in the spleen. GRB2, which is highly associated with DR in T2D patients, is likewise essential for pro-B cell survival and maturation, with its absence depleting newly mature B cells in the spleen. Overall, splenic B-cell abundance could serve as a predictive biomarker for DR risk.

Cis-eQTL analysis in thyroid tissue demonstrated that the allelic configuration of rs6066146 significantly modulates EYA2 expression, and thyroid tissue also showed nominal significance in our TWAS. Although we did not uncover direct pathway evidence linking thyroid to DR in our study, several studies support a clinical connection. Subclinical hypothyroidism is associated with an increased risk of sight-threatening DR, and in euthyroid T2D patients, free triiodothyronine levels within the normal range are negatively associated with DR. Moreover, thyroid dysfunction alters VEGF levels, a key driver of retinal angiogenesis and permeability, and elevated VEGF in Graves’ disease and Hashimoto’s thyroiditis may exacerbate DR progression.”

Rather than testing 48 GTEx tissues, we chose to restrict our TWAS to the spleen and thyroid. We believe a priori-driven TWAS focused on the spleen and thyroid is most appropriate, as these tissues were independently nominated by our FUMA-based tissue-expression and cis-eQTL screens. This targeted approach reduces the multiple-testing burden inherent in a 48-tissue scan and increases power to detect relevant associations. Therefore, our TWAS was restricted to these two tissues, and, with only two tests for the *EYA2* module, not for the all-genes module in chromosome 20. We report nominal significance at $p < 0.05$ without Bonferroni correction. These details are now included in the Methods (lines 139-142) and Results (line 255) sections.

We think it is not necessary to test all tissues as most tissues lack evidence linking with DR. However, if the reviewer considers it necessary, we are happy to extend our analysis to all GTEx tissues in a full TWAS.

Methods: “Rather than testing 48 GTEx tissues, our analysis was limited a priori to the specific tissues identified by prior tissue-expression and cis-eQTL screening, and only tests for gene modules harboring significant SNPs. Because this analysis was a priori-driven, we report nominal significance at $p < 0.05$ without additional multiple-testing correction.”

Results: “TWAS for EYA2 was restricted to the spleen and thyroid based on prior screens.”

5. Regarding the Mendelian Randomization (MR) analysis:

- State the total number of ocular traits examined.
- Justify the selection of "Eye or Eyelid Problems" and glaucoma as exposures for the analysis.

Response: Thanks for your valuable insights. We totally evaluated three ocular-related trait categories—cataract (samples $n=2$), glaucoma ($n=2$), and eyelid problems ($n=5$)—because each represents a potential diabetic eye comorbidity that may be related to DR (PMID: 27917530 [glaucoma & cataract]; PMID: 39329649 [eye or eyelid problems]), all derived from European GWAS summary statistics (see Supplementary Table 4 for details). We then conducted bidirectional MR to explore their associations without specifying any particular trait as exposure or outcome. We have revised the Methods (lines 145-149) section to provide detailed descriptions.

In this revision, given that proinsulin alterations in early T2D precipitate DR (<https://doi.org/10.1134/S0022093017030024>), we also incorporated genetic instruments for type 2 diabetes ($n=41$) and proinsulin levels ($n=1$) in both univariable (bidirectional) and multivariable MR analyses to disentangle their individual and joint causal effects on DR.

Methods: “We conducted bidirectional MR analyses to assess potential causal relationships between DR and T2D-related traits. Specifically, we incorporated genetic instruments for type 2 diabetes (tests samples $n=41$) and proinsulin levels ($n=1$) given that proinsulin alterations in early T2D precipitate DR, and for cataract ($n=2$), glaucoma ($n=2$), and eyelid disorders ($n=5$) since these ocular diseases are related to DR.”

6. Discuss whether any known DR-associated loci were replicated in this GWAS or if they appeared at least nominally significant, providing more detailed insights into the consistency of these findings with previous studies.

Response: Thank you for your suggestion. We looked up top DR-associated SNPs from five independent cohorts/publications in our UKB DR GWAS (Supplementary Table 5). Three loci showed at least nominal replication ($p < 0.05$): African American ancestry rs7903146 ($p=0.039$)

and rs2237897 ($p=0.037$), and European ancestry rs34872471 (0.049). The Chinese GWAS SNP rs1399634 was borderline ($p=0.050$). Other variants did not replicate ($p > 0.05$).

This modest replication rate likely reflects differences in phenotype definition (ICD-10 codes in UKB vs. PheCode or DR grading elsewhere), ancestry heterogeneity, and variable case-ascertainment protocols across cohorts. Nonetheless, the nominal signals at well-established loci such as rs7903146 and rs34872471 support their relevance to DR. We have added this in the Results (lines 197-201) and Discussion (lines 385-388) to present these replication findings and pinpoint the complexities of cross-cohort consistency in DR genetics.

Results: “We further assessed the replication of top DR-associated SNPs reported in five independent cohorts within our UKB DR GWAS. Three loci showed nominal replication ($p < 0.05$): the African American ancestry SNPs rs7903146 ($p=0.039$) and rs2237897 ($p=0.037$), and the European-ancestry SNP rs34872471 ($p=0.049$). The Chinese GWAS SNP rs1399634 was borderline significant ($p=0.050$), while the remaining variants did not replicate (Supplementary Table 5).”

Discussion: “Moreover, when we looked up other top DR-associated SNPs from discovery cohorts in our GWAS summary statistics, three showed nominal replication, meaning that differences in phenotype definitions, ancestral heterogeneity, and variable case-ascertainment protocols did not detract from the substantive insights (Supplementary Table 5).”

7. The inclusion of the FinnGen cohort might introduce bias due to its use of population controls. Please discuss the implications of this for the genetic association results.

Response: Thank you for your valuable insights regarding the impact of FinnGen’s control definitions. As noted in Answer (1), as a replication cohort, FinnGen’s use of population controls may introduce bias by mixing diabetes-related and DR-specific signals, thereby increasing the type I error rate in our replication study. We discussed this in lines 376-381.

Discussion: “Another weakness of our study is in the replication stage. We used cohorts such as FinnGen and similarly the African American and European, which contain population controls rather than diabetes-specific controls in their GWAS on DR. This study design may elevate type I error, since true signals can fail to replicate in less well-defined replication cohorts. Fortunately, our downstream analyses demonstrate that the rs6066146 variant influences both T2D and DR, and that, within the EYA2 region, T2D predisposes to DR. However, it is still recommended that the results generated should be interpreted with caution.”

8. Given that variants in *EYA2* have been previously associated with type 2 diabetes (T2D) in other studies (Nat Genet. 2018;50(11):1505–1513. doi: 10.1038/s41588-018-0241-6), highlight the evidence supporting the specificity of *EYA2*'s involvement in DR, distinguishing it from general T2D susceptibility.

Response: Thank you for your insightful comment. We acknowledge that *EYA2* has been linked to T2D susceptibility. First, unlike standard T2D GWAS, our discovery analysis compared DR cases to non-DR controls drawn exclusively from T2D patients in UKB, and *EYA2* emerged as the only genome-wide significant locus, underscoring its specific association with retinopathy rather than general diabetes risk. Second, we performed Bayesian colocalization within ± 500 kb of rs6066146 (identified from our UK Biobank DR GWAS) between DR GWAS summary statistics and a large T2D meta-analysis (Nat Genet 2018;50(11):1505–1513). This analysis yielded $PP.H3 = 0.433$ (distinct causal variants) and $PP.H4 = 0.553$ (shared causal variant), indicating the presence of both shared and independent association signals of T2D and DR at the *EYA2* locus. Third, cis-MR provided further evidence that genetically driven T2D variation at this gene exerts a significant causal effect on DR. Together, these results demonstrate a DR-specific effect at the *EYA2* locus beyond its role in T2D susceptibility.

We added and discussed these in the Results (lines 269-272 & 291-294) and Discussion (lines 302-306) to demonstrate a DR-specific signal at *EYA2* distinct from its general T2D association.

Results: “A subsequent cis-MR at the *EYA2* locus, where the gene-T2D association in ExPheWas is $p=3.0 \times 10^{-7}$, estimated an IVW odds ratio of 2.52 (95% CI: 1.22-5.21) for T2D on DR ($p=0.013$), providing robust, gene-centric evidence of a T2D-DR causal pathway (Supplementary Fig. 5).”

“Using T2D summary statistics from Mahajan et al. (2018) and our UKB DR GWAS at the *EYA2* locus, colocalization yielded $PP.H3 = 0.433$ and $PP.H4 = 0.553$, reflecting moderate support for a shared causal variant alongside independent effects, though $PP.H4$ falls below the conventional $>80\%$ threshold for definitive colocalization (Supplementary Table 13 and Supplementary Fig. 7).”

Discussion: “Notably, *EYA2* has been identified as an active promoter for T2D and also emerges as a key driver of DR. Our colocalization analysis at the *EYA2* locus provided suggestive evidence of a shared causal variant ($PP.H4 = 0.553$) alongside independent association signals, and subsequent cis-MR provided further evidence that genetically driven T2D variation at this gene exerts a significant causal effect on DR.”

Reviewer #2 (Remarks to the Author):

The authors conducted a GWAS using the UK Biobank and identified a genome-wide significant association between diabetic retinopathy (DR) and the *EYA2* gene variant rs6066146 in type 2 diabetes patients. The association was replicated in the FinnGen cohort, and functional analyses suggest involvement of immune and angiogenic pathways. Mendelian randomization further identified glaucoma and eyelid problems as potential risk factors for DR. Please see the specific comments below for details.

1. While DR cases were identified using ICD-10 code H360, it is unclear whether individuals with other retinal diseases (e.g., age-related macular degeneration) were excluded. Please clarify whether such conditions were considered and how misclassification risk was managed.

Response: Thank you for raising this point. Our discovery GWAS was confined to individuals with type 2 diabetes, intentionally allowing diabetes-related ocular comorbidities to remain in the case definition; excluding every co-occurring retinal disorder would have reduced the original case (1,731) and markedly lowered statistical power. To assess possible misclassification by age-related macular degeneration (AMD), we conducted a sensitivity GWAS that removed all participants carrying the ICD-10 code H35.3. This excluded 270 individuals ($\approx 15\%$ of DR cases).

*After AMD exclusion, rs6066146 at the *EYA2* locus remained suggestive ($p=5.8\times 10^{-6}$) with $\text{Beta(SE)} = 0.015(0.003)$. The modest loss of significance likely reflects the high co-prevalence of AMD and DR in older patients ($>90\%$ of AMD occurs after age 55); removing AMD cases, therefore, also eliminates older DR cases and slightly reduces power. Importantly, the persistence and direction of the DR signal indicate that any residual AMD comorbidity in our T2D-restricted sample does not materially influence the association.*

We further queried rs6066146 in external resources (e.g., T2D Knowledge Portal) and found no evidence of association with AMD or other non-DR ocular traits, reinforcing its specificity for DR. These sensitivity results have been added to the Discussion (lines 367-375).

In summary, we have considered such conditions in our GWAS. If a person has DR and AMD, we consider he/she as a DR case regardless of the AMD status.

*Discussion: "Since our GWAS analysis included DR cases with comorbid AMD and other retinal diseases, we acknowledge that we sacrificed some phenotypic purity but gained more case samples, thereby increasing statistical power. In sensitivity analyses excluding 270 AMD cases ($\approx 15\%$ of the original 1,731 DR cases), the *EYA2* variant rs6066146 remained suggestively associated with DR ($p=5.8\times 10^{-6}$; Supplementary Fig. 8) with $\text{Beta(SE)} =$*

0.015(0.003). The slight attenuation of significance likely reflects the high co-occurrence of AMD and DR in older individuals (Supplementary Fig. 9), as removing AMD cases also disproportionately removes older DR cases and reduces power. Crucially, the persistence and consistent direction of the DR-association demonstrate that residual AMD comorbidity in our T2D-restricted sample does not materially affect the validity of the EYA2-DR signal.”

2. A more detailed comparison of the discovery and replication cohorts—particularly regarding phenotype definitions and disease ascertainment—would help readers better understand the replication results.

Response: Thank you for this suggestion. We have added a detailed comparison of all cohorts to Supplementary Table S1, including ancestry, case definitions, control definitions, and ascertainment methods. Notably, the GoDARTS replication, like our discovery GWAS, restricted both cases and controls to individuals with diabetes mellitus, thereby eliminating diabetes status as a confounder; however, GoDARTS also included a subset of type 1 diabetes, which likely reduced power to replicate our confirmed T2D-specific DR association. In contrast, FinnGen, African American, and European ancestry cohorts used population controls, retaining diabetes-mediated genetic variation among study and enhancing sensitivity to detect DR loci, although this may elevate the type I error rate. Their successful replication further underscores the robustness of our findings. We have underscored this in the Discussion (lines 376-388).

Discussion: *“Another weakness of our study is in the replication stage. We used cohorts such as FinnGen and similarly the African American and European, which contain population controls rather than diabetes-specific controls in their GWAS on DR. This study design may elevate type I error, since true signals can fail to replicate in less well-defined replication cohorts. Fortunately, our downstream analyses demonstrate that the rs6066146 variant influences both T2D and DR, and that, within the EYA2 region, T2D predisposes to DR. However, it is still recommended that the results generated should be interpreted with caution. For the GoDARTS study, although using the diabetes controls (Supplementary Table 2), it also included a subset of T1D cases, which reduced power to replicate our T2D-specific DR signal. Despite these limitations, the consistent replication ($p < 0.05$ & $BETA > 0$) of rs6066146 across three cohorts supports its robustness across ancestries. Moreover, when we looked up other top DR-associated SNPs from discovery cohorts in our GWAS summary statistics, three showed nominal replication, meaning that differences in phenotype definitions, ancestral heterogeneity, and variable case-ascertainment protocols did not detract from the substantive insights (Supplementary Table 5).”*

3. Please provide more details on key parameters used in TWAS and Mendelian randomization analyses, such as SNP clumping thresholds, eQTL reference panels, and multiple testing correction approaches.

Response: Thank you for the suggestion. We have expanded the Methods section to include detailed TWAS and MR parameters (lines 134-142 & 149-159).

Methods: *“We employed Transcriptome-Wide Association Studies (TWAS) (<http://gusevlab.org/projects/fusion/>) to assess the impact of gene expression modulated by genetic variants on DR risk, with expression weights derived from the GTEx v7 tissue panel and utilized 1000 Genomes Phase 3 European (1000G.EUR) haplotypes as our LD reference. We ran FUSION’s default suite of predictive models (BLUP, LASSO, etc.) and selected the best-performing weight set per gene based on cross-validation R^2 . Our analysis was limited a priori to the specific tissues identified by prior tissue-expression and cis-eQTL screening, and only tests for gene modules harboring significant SNPs. Because this analysis was a priori-driven, we report nominal significance at $p < 0.05$ without additional multiple-testing correction.*

These analyses were performed using the TwoSampleMR package in R, with all summary statistics derived from the European GWAS and accessed via the IEU OpenGWAS database (Ben Elsworth et al.; see Supplementary Table 4 for details). For each exposure, we selected independent instruments meeting genome-wide significance ($p < 5 \times 10^{-8}$) and applied clumping ($r^2 < 0.001$ within a 10,000 kb window) against the 1000 Genomes Phase 3 European reference. Variants were then harmonized across exposure and outcome datasets, excluding ambiguous palindromic SNPs. Multiple MR estimators were applied, including inverse variance weighted (IVW), MR Egger, weighted median, simple median, and weighted mode. Cochran’s Q test was used to assess heterogeneity, and the MR-Egger intercept test for horizontal pleiotropy. Using MR-PRESSO to detect and correct horizontal pleiotropy by removing outliers. As our analyses were confined to hypothesis-driven trait pairs, we report nominal $p < 0.05$ without further multiple-testing correction.”

4. Since the GWAS was limited to a white British population, a brief discussion of how findings may or may not generalize to other ancestries would be valuable.

Response: Thank you for pointing this out. We have added a sentence in the Discussion (lines 388-391) noting that, although our lead signal replicated in Finnish and African American cohorts, most of the follow-up analyses (e.g., Mendelian Randomization, colocalization) relied on European LD panels. Therefore, in future studies, we will carefully consider this and may study non-European populations to confirm the transferability of the *EYA2* association.

Discussion: *“Since our study is limited to the white British population and most of the downstream analyses used European LD panels, additional studies in non-European populations are still required to confirm the generalizability of the EYA2 association.”*

5. The manuscript reports spleen tissue enrichment and a B cell–related gene set, but the connection between the two is not discussed. Given the spleen’s role in B cell biology, a brief interpretation of this potential link may strengthen the discussion.

Response: Thank you for this valuable suggestion. We have now expanded our Discussion section (lines 322-330) to explicitly link spleen enrichment with B-cell biology. Specifically, the significant gene set "GOBP_REGULATION_OF_PRO_B_CELL_DIFFERENTIATION," driven primarily by *SOS1* and *SOS2*, is consistent with our tissue-level analyses highlighting spleen enrichment. Relevant literature (PMID: 24043312; PMID: 21427701) further supports this connection, demonstrating reduced splenic mature B-cell counts in *SOS1/2* double-knockout mice and impaired B-cell maturation following *Grb2* knockout, a gene strongly linked to DR in T2D. Collectively, these results suggest that alterations in splenic B-cell populations may reflect important biological pathways underlying DR pathogenesis.

Discussion: *“The gene set GOBP_REGULATION_OF_PRO_B_CELL_DIFFERENTIATION reached genome-wide significance in our MAGMA analysis, with SOS1 and SOS2 driving this signal. Consistent with this, the spleen emerged as a significant tissue in both our tissue expression and chromatin-interaction analyses (the loci at chr20:45,560,001-45,600,000 and 45,600,001-45,640,000 showed strong interactions), and received nominal support from TWAS. These findings align with murine studies demonstrating the relationship between the spleen and B cells. SOS1/2 double-knockout mice exhibit a significant reduction of mature B cells in the spleen. GRB2, which is highly associated with DR in T2D patients, is likewise essential for pro-B cell survival and maturation, with its absence depleting newly mature B cells in the spleen. Overall, splenic B-cell abundance could serve as a predictive biomarker for DR risk.”*

In summary, we detailed all methods and incorporated additional strategies to strengthen the rigor of our results. We sincerely thank the reviewers and the editor once again for the valuable comments and would be happy to consider any further suggestions you may have.

Best wishes,

Dr Weihua Meng and co-authors

Dear Editors and Reviewers:

We sincerely thank you for the time and effort you have devoted to evaluating our manuscript. Your insightful comments and suggestions have been invaluable in helping us strengthen the clarity, rigor, and overall quality of the work. We have revised the manuscript carefully in line with the editorial checklist.

Reviewers' comments:

Reviewer #1 (Remarks to the Author):

Thank you for your responses. The authors have successfully addressed most of my concerns.

Response: We appreciate the reviewer's constructive feedback and are pleased that our revisions have addressed the majority of the concerns raised.

In summary, we have updated the figure and table legends to provide complete statistical details and have verified that all essential author information (e.g., author names, corresponding authors' ORCIDs and email addresses) is accurate and complete. We also revised the Plain Language Summary and provided a draft Editorial Summary as requested.

We again thank the editors and reviewers for their helpful comments. Please do not hesitate to contact us if you require any additional information.

Best wishes,

Dr Weihua Meng and co-authors